# Genomic Selection Using Single-Step Genomic BLUP on the Number of Services per Conception Trait in Thai–Holstein Crossbreeds

**DOI:** 10.3390/ani13233609

**Published:** 2023-11-22

**Authors:** Wuttigrai Boonkum, Vibuntita Chankitisakul, Monchai Duangjinda, Sayan Buaban, Pattarapol Sumreddee, Piriyaporn Sungkhapreecha

**Affiliations:** 1Department of Animal Science, Faculty of Agriculture, Khon Kean University, Khon Kean 40002, Thailand; vibuch@kku.ac.th (V.C.); monchai@kku.ac.th (M.D.); 2Network Center for Animal Breeding and Omics Research, Khon Kaen University, Khon Kaen 40002, Thailand; 3Bureau of Animal Husbandry and Genetic Improvement, Department of Livestock Development, Pathum Thani 12000, Thailand; buaban_ai@hotmail.com; 4Bureau of Biotechnology in Livestock Production, Department of Livestock Development, Pathum Thani 12000, Thailand; pattrapoljk@gmail.com; 5Department of Animal Science, Faculty of Agricultural Technology, Rajamangala University of Technology Thanyaburi, Pathum Thani 12110, Thailand; piriyaporn_s@rmutt.ac.th

**Keywords:** accuracy, fertility, ssGBLUP, tropical area, Thai dairy cattle

## Abstract

**Simple Summary:**

The development of dairy production systems has continued to progress, especially in purebred dairy cows in the United States and Europe. However, in-depth knowledge is still needed to support decision-making for crossbreed dairy cows raised in Asia to be used in breeding programs. The development of a new genetic method that is more effective than the current method is what everyone wants. Therefore, to ensure that the new method can replace the current method, it must be tested with the utmost confidence.

**Abstract:**

Reproductive traits are important traits that directly affect a farmer’s income and are difficult to improve upon using traditional genetic methods. Therefore, there is a need to consider new options for increasing the accuracy of the genetic selection of dairy cows. The objective of this study was to compare the genetic methods of the traditional BLUP and ssGBLUP techniques in terms of the estimated genetic parameters and accuracy of the estimated breeding values. The data comprised 101,331 services per conception (NSPC) records from 54,027 Thai–Holstein crossbred cows, 109,233 pedigree data, and 770 genotyped animals. A Bayesian analysis via threshold Gibbs sampling was used to analyze the estimated variance components and genetic parameters. The results showed that the means of the NSPC data were 2.21, 2.31, and 2.42 for less than 87.5% for Holstein genetics (breed group; BG1), 87.5 to 93.6% for Holstein genetics (BG2), and greater than 93.7% for Holstein genetics (BG3), respectively. The estimated heritability values were 0.038 and 0.051, and the repeatability values were 0.149 and 0.157 for the traditional BLUP and ssGBLUP methods, respectively. The accuracy of the estimated breeding values from the ssGBLUP method was higher than that from the traditional BLUP method, ranging from 6.05 to 17.69%, depending on the dataset, especially in the top 20% of the bull dataset had the highest values. In conclusion, the ssGBLUP method could improve the heritability value and accuracy of the breeding values compared with the traditional BLUP method. Therefore, switching from traditional methods to the ssGBLUP method for the Thai dairy cattle breeding program is a viable option.

## 1. Introduction

Reproductive characteristics are essential to livestock and dairy production and are also related to the productivity of dairy cows with a direct correlation to the cost and profits of farmers [1,2]. However, it is evident that negative genetic correlations between milk production and reproductive performance have resulted in a decline in dairy cow fertility [3,4]. In addition, a negative correlation exists in cattle for reproductive traits when the temperature and humidity index (THI) crosses a threshold level or when heat stress occurs [5]. In Thailand, most dairy cows are crossbred between *Bos taurus* (Holstein) and *Bos indicus* (Sahiwal, Brahman, Thai native) and are selected primarily for the purpose of increasing milk yield and milk composition [6,7]. However, a negative effect in terms of poor fertility was reported thereafter. For example, the calving season influenced the days open of the Thai–Holstein crossbreeds. In addition, a decrease in superovulation and a higher incidence of degenerated embryos were found in superovulated cows exposed to heat stress [8]. Therefore, improving the reproductive performance of dairy cows is necessary. An increasing number of inseminations is one of the main parameters that characterize low fertility. The number of services per conception (NSPC) is related to cost, and expanding the day service decreases the fertility performance and reduces the production performance and economic value. The NSPC problems are related to increasing days open, conception, and nonreturn rates. The higher numbers for NSPC show increasing AI service techniques and accuracy in checking the date to service [9]. In Thailand, Thai–Holstein crossbreeds had an average number of services per conception of 1.55–2.29 times [3,10], while those of other countries were 1.70 times in Holstein–Friesian cows from Japan [11] and 1.20–2.37 times in crossbred dairy cows in central Ethiopia [12,13]. The NSPC is a problem mainly on small-scale farms in tropical areas, focusing on Thailand.

The heritability of fertility traits in dairy cows is generally low because the environmental factors, feed and feeding effects, and management influences are more relevant than genetic effects [14,15]. Rahbar et al. [16] found that heritability ranged from 0.02 to 0.12, suggesting little genetic influence. For the NSPC in Thailand, Buaban et al. [3] used the traditional method to evaluate and show low heritability (0.02), which also affected the low selection accuracy. Other dairy cow populations also have low heritability, such as Japanese black cattle (0.04 to 0.22) [17], Holstein–Friesian cows in Ireland (0.02) [18], and Danish Holstein populations (0.03) [9]. Therefore, it is challenging for breeders to find a method to improve the fertility traits in breeding programs. One possible solution for increasing the selection accuracy is based on the genomic selection method proposed by Meuwissen et al. [19], which uses phenotype, pedigree, and genotype data to evaluate the genomic estimated breeding values (GEBVs). Misztal et al. [20] proposed that the genomic breeding values could be obtained using single-step genomic selection BLUP (ssGBLUP). The single-step BLUP method combines information from pedigree relationships and genomic information (marker data) in a single analysis. The “single step” aspect integrates the pedigree and genomic information in a unified analysis. The traditional BLUP methods separately analyze the pedigree and genomic data and then combine the results. On the other hand, the single-step BLUP considers both sources of information simultaneously, providing more accurate and efficient estimates [21,22,23]. The ssGBLUP method can improve genetics, increase accuracy, increase genetic gain, and reduce generation intervals [24,25,26]. However, the previous reports have only studied purebred dairy populations. There have yet to be any reports in crossbred dairy populations in which the same results were obtained. Therefore, the objective of this study was to estimate the genetic parameters and compare the accuracy analysis of the estimated breeding value between the traditional BLUP method and genomic selection using the single-step genomic BLUP method. Hopefully, the benefits from this study will be considered to adjust the method in crossbred dairy cattle breeding programs in tropical areas, including Thailand.

## 2. Materials and Methods

### 2.1. Data Collection

The Institutional Animal Care and Use Committee of Khon Kaen University reviewed and approved this research based on the Ethics of Animal Experimentation of the National Research Council of Thailand (No. IACUC-KKU-120/64, 30 November 2021). A total of 101,331 services per conception (NSPC) records for the first fifth lactation of 54,027 Thai–Holstein crossbred cows collected between 1996 and 2017 were provided by the Bureau of Biotechnology in Livestock Production, Department of Livestock Development, Thailand. The NSPC data were defined as the outcome of all the insemination events, with “1” being a successful conception and an unsuccessful service conception event being “0”. The cows with ambiguous identification for the date of each record, identity data, and more than one successful insemination per parity or an unrealistic interval between consecutive inseminations were removed. The pedigree data included 109,233 animals born between 1994 and 2016, and 770 animals were genotyped using the Illumina BovineSNP50 Bead Chip (Illumina Inc., San Diego, CA, USA). Quality control on the animals and markers was performed according to the following parameters: a minimum call rate equal to 90% and a minor allele frequency for each marker greater than 5%. The animals and markers that failed these quality control criteria were removed. The cows were grouped by the percentage of Holstein genetics (breed group; BG) as follows: BG1 less than 87.5%, BG2 from 87.5 to 93.6%, and BG3 greater than 93.7%. The age at the first service was divided into seven classes at 3-month intervals, with less than 25 months comprising the first class and greater than 39 months comprising the last. The details of the descriptive statistics of the studied traits are shown in Table 1.

### 2.2. Statistical and Genetic Analysis

The number of services per conception data (NSPC) were validated and analyzed for the least square means, and significant differences were compared by breed group and parity using the *Scheffe’* (*p* < 0.05) post hoc test in the generalized linear model for an unbalanced analysis of variance (GLM procedure) using the SAS package. The genetic analysis used the repeatability threshold animal model to estimate the variance components, heritability, and breeding value. The outcomes of the inseminations after calving during the breeding period were considered to be repeated observations. Therefore, the animal’s additive genetic effects (a), permanent environmental effects (p) and residual effects (e) were modeled as constants. The model equation was written as follows.
y=Xβ+Zhy+Za+Zp+Zss+e
where y is the vectors of unobserved liabilities for the NSPC records from 1 to 10; β is a vector of the fixed effects (year–month of insemination, age at the first service group, breed group, and parity); hy is a vector of the contemporary group (CG) random herd-year of the insemination effects; a is a vector of the random additive genetic effects; p is a vector of the random permanent environmental effects; ss is a vector of the random effects from the service sire effects; e is a vector of the random residual effects; X is the incidence matrix for the fixed effects; and Z is the incidence matrix for the random effects. The covariance structures were assumed as follows.
$$\mathrm{Var}\;\left[\begin{array}{c} hy\\ a\\ p\\ ss\\ e \end{array}\right]\;=\left[\begin{array}{ccccc} I\sigma_{hy}^{2} & 0 & 0 & 0 & 0\\ 0 & A\sigma_{a}^{2} & 0 & 0 & 0\\ 0 & 0 & I\sigma_{p}^{2} & 0 & 0\\ 0 & 0 & 0 & I\sigma_{ss}^{2} & 0\\ 0 & 0 & 0 & 0 & I\sigma_{e}^{2} \end{array}\right]$$
where A is the additive relationship matrix among the animals, I is the identity matrix, and $\sigma_{hy}^{2}$, $\sigma_{a}^{2}$,$\;\sigma_{p}^{2}$, $\sigma_{ss}^{2}$, and $\sigma_{e}^{2}$ are the herd-year of insemination, additive genetics, permanent environment, service sire, and residual variances, respectively.

The variance components, genetic parameters (heritability, repeatability), estimated breeding values (EBVs), and genomic estimated breeding values (GEBVs) were estimated using a Bayesian implementation via Gibbs sampling. The computations were carried out using the THRGIBBS1F90 program [27]. The number of iterations was set to 500,000. The first 50,000 samples were discarded as burn-in, and every 10th sample was kept thereafter. The post-Gibbs analysis using the POSTGIBBSF90 program [27] was conducted to obtain the posterior distribution statistics and verify the parameter estimates.

### 2.3. Estimation of the Genetic Parameters

The estimated heritability $\left(h^{2}\right)$ and repeatability $\left(r\right)\;$in both the BLUP and ssGBLUP methods were defined as the following.
$$h^{2}=\frac{\sigma_{a}^{2}}{\sigma_{hy}^{2}+\sigma_{a}^{2}+\sigma_{p}^{2}+\sigma_{ss}^{2}+\sigma_{e}^{2}}\;r=\frac{\sigma_{a}^{2}+\sigma_{p}^{2}}{\sigma_{hy}^{2}+\sigma_{a}^{2}+\sigma_{p}^{2}+\sigma_{ss}^{2}+\sigma_{e}^{2}}$$
where $\sigma_{hy}^{2}$ is the herd-year of the insemination variance,$\;\sigma_{a}^{2}$ is the additive genetic variance, $\sigma_{p}^{2}$ is the permanent environmental variance, $\sigma_{ss}^{2}\;$is the service sire variance, and $\sigma_{e}^{2}$ is the residual variance. The goodness of fit used to compare the methods was the lowest −2logL (negative two log-likelihood) and DIC (deviance information criterion). The accuracies $\left(acc\right)$ of the EBVs from the BLUP method and the GEBVs from the ssGBLUP method were calculated using the theoretical accuracy method with the following equation$\;acc=\sqrt{1-\left(\frac{SE^{2}}{\sigma_{a}^{2}}\right)}$, where $SE^{2}$ is the prediction error variance and $\sigma_{a}^{2}$ is the additive genetic variance [28].

### 2.4. Estimation of the Breeding Values

Genomic analyses were implemented using a single-step genomic method. The inverse of the numerator relationship matrix (A) in the mixed model equations was replaced by the inverse of the matrix H [21], as follows.
$$H^{-1}=A^{-1}+\left[\begin{array}{cc} 0 & 0\\ 0 & G^{-1}-A_{22}^{-1} \end{array}\right]$$
where $A^{-1}$ is the inverse of a pedigree-based relationship matrix for all the animals included in the analysis, $A_{22}^{-1}$ is the inverse of the pedigree-based relationship for the genotyped animals only, and $G^{-1}$ is the inverse of a genomic relationship matrix (G), constructed similar to VanRaden [29]. The estimated breeding values (EBVs) and genomic breeding values (GEBVs) were analyzed by the BLUPF90 program [30] with the posterior means of the (co)variance components at their estimated values, as follows.
$$\begin{array}{l} \mathrm{EBV}=\;\hat{a}\\ \mathrm{GEBV}=\hat{a_{i}}+\sum {\mathrm{SNP}}_{i} \end{array}$$

## 3. Results

### 3.1. Comparison of the Number of Services per Conception between the Breed Groups within Parity

A comparison of the average number of services per conception (NSPC) separated by parity and breed group in Thai–Holstein crossbreds is presented in Figure 1. The mean values of the NSPC characteristics increased with the genetic percentage of the Holstein breed and lactation. During the first and second lactations, significant differences were not found (*p* > 0.05). However, there was a significant difference from the third lactation onward (*p* < 0.05), and the difference was more pronounced from the fourth and fifth lactations. The mean value of the NSPC characteristics in the BG3 group (greater than 93.7% Holstein genetics) was the highest, while those in the BG1 group (less than 87.5% Holstein genetics) were the lowest.

### 3.2. Variance Components and Genetic Parameters

A comparison between the variance components and genetic parameters of the NSPC traits between the BLUP and ssGBLUP methods is presented in Table 2. The additive variance in the ssGBLUP method (0.12) was higher than that in the BLUP method (0.09), while the permanent environmental and residual variances in the ssGBLUP method (0.25 and 1.80) were slightly lower than those in the BLUP method (0.26 and 1.82). The estimated heritability was 0.038 for the BLUP method and 0.051 for the ssGBLUP method. At the same time, the repeatability was higher than the heritability, and was 0.149 for the BLUP and 0.157 for the ssGBLUP methods. The statistical criteria showed that both the DIC and −2logL values of the ssGBLUP method were lower than those of the BLUP method.

### 3.3. Comparison of the Accuracy between the EBVs and GEBVs

The accuracies of the estimated breeding values (EBVs) and genomic estimated breeding values (GEBVs) from the BLUP and ssGBLUP methods are presented in Table 3. The datasets were compared in several scenarios: all datasets, the dam dataset, the bull dataset, top 20% of all the datasets, top 20% of the dam dataset, and top 20% of the bull dataset. The results showed that the accuracy of the GEBVs from the ssGBLUP method (0.490, 0.498, 0.521, 0.560, 0.572, 0.612) was higher than the accuracy of the EBVs from the BLUP method (0.462, 0.465, 0.468, 0.504, 0.508, 0.520) in all the datasets, especially in the 20% of the bull dataset, which had the highest accuracy compared to the other datasets. In addition, when compared as the percentage of differences, it was found that the values were in the range of 6.05 to 17.69%.

### 3.4. Difference between the EBVs and GEBVs

The average estimated breeding values (EBVs) from the BLUP method and genomic estimated breeding values (GEBVs) from the ssGBLUP method for the number of services per conception separated by dataset and breed group are presented in Table 4. In general, an animal with a negative breeding value for the number of services per conception (NSPC) indicated that the animal was genetically good in insemination with fewer insemination times compared to the breeding value of the herd average. The results from this study showed that the GEBVs from the ssGBLUP method were more negative than the EBVs from the BLUP method in all the datasets, especially in the top 20% the bull dataset, which had the highest negative values. The most negative values were found in the BG3 group of the Thai–Holstein crossbreeds, followed by the BG2 and BG1 groups. In the top 20% of the bull dataset, the GEBVs in BG3, BG2, and BG1 were −0.185, −0.176, and −0.165, while the EBVs were −0.180, −0.172, and −0.160, respectively.

## 4. Discussion

The number of services per conception (NSPC) is a fertility trait in dairy cows related to reproductive performance and the cost of management in artificial insemination service. In this study, the average values of the NSPC in the Thai–Holstein population ranged from 2.10 to 2.52 times. Compared to what was reported in the same dairy population in 2015 [3,10] and 2016, the NSPC was found to be 10–35% higher. It was also higher than that in other populations, including Holstein dairy cattle in the UK and Ireland (1.94 ± 1.29 times) [31], Russian black-and-white cattle population (1.80 ± 1.39 times) [32], and crossbred dairy cows in central Ethiopia (1.59 ± 1.02 times) [13]. The results from this study indicated that Thai–Holstein cows have significantly lower fertility, especially in cows with high Holstein genetics greater than 93.7%. Wattiaux [33] suggested that the optimal value of the NSPC should be lower than 1.70 times, and if the NSPC value is higher than 2.50 times, dairy herds are at risk of reproductive problems. The NSPC has become the result of artificial insemination, reduced production and lifetime productivity, increased culling rates, and the replacement costs in dairy farms [9,34,35]. Therefore, as an initial solution to prevent excessively high NSPC values, farmers may consider raising crossbred dairy cows whose Holstein genetics are lower than 93.7%.

The estimated heritability values of the NSPC from the BLUP and ssGBLUP methods were 0.038 and 0.051, respectively. These heritability values were higher than those of the same population in 2015, which were 0.02–0.03 [3,10]. In addition, there were also higher values than those reported in Holstein–Friesian cows in Ireland at 0.02 [18], in Danish Holstein populations at approximately 0.03 [9], in UK and Ireland Holstein dairy cattle ranging from 0.016 to 0.019 [31] and in Spanish Holstein cows at 0.038 [36]. Additionally, there were lower values in some reports, such as in Russian black-and-white cattle populations at 0.11 [32] and Italian Brown Swiss cattle at 0.061–0.067 [37]. However, the heritability of this study was in the same range as those in Japanese black cows, ranging from 0.04 to 0.22 from parity 1 to 10 [17], and those in Holstein and Jersey cows of Colombia at 0.04 and 0.09, respectively [38]. Although the heritability from this study showed small values, it still increased when compared to that of the same population in 2015, which means that the effect of the animal genetic selection program and the phenotype was controlled by the effect of increased genetic variation. In addition, the heritability estimated by the ssGBLUP method was higher than that estimated by the BLUP method because the additive genetic variance estimated by the ssGBLUP method was higher and the residual variance was lower than those estimated by the BLUP method. 

The estimated additive genetic variance from the ssGBLUP method is often higher than that from the traditional BLUP method for the following reasons. The ssGBLUP method incorporates genomic information, such as single-nucleotide polymorphism (SNP) markers, which directly measure an individual’s genetic makeup. By leveraging genomic data, the method can capture the genetic variations that are not accounted for by pedigree information alone. Therefore, the increased accuracy results in a better estimation of the additive genetic variance. The ssGBLUP method can handle incomplete pedigrees more effectively by integrating genomic data, which helps to compensate for the missing or uncertain relationships between the individuals. Additionally, the ssGBLUP method can account for some of these nonadditive genetic effects by using genomic data, leading to a higher estimated additive genetic variance. Finally, the ssGBLUP method can utilize linkage disequilibrium (LD) patterns between the markers and causal genes. LD refers to the nonrandom association of alleles at different loci, which can infer the effects of specific genes even if they are not directly genotyped. This extra information contributes to a more accurate estimation of the additive genetic variance. 

Although the ssGBLUP method increases the overall heritability value, in the expression of the NSPC in dairy cows, it was found that most of the factors that control the expression of these traits were environmental factors rather than genetic [39]. For this reason, improving fertility traits can be achieved by improving reproductive management, including finding effective estrus methods, artificial insemination at the right time, avoiding heat-stressed cattle, ensuring qualified artificial insemination technicians are used, and that they use effective hygiene before and after artificial insemination. In addition, proper diet management for both heifers and postpartum cows should be performed in association with the health of the reproductive system [15,16].

We found that the accuracy of the GEBVs from genomic selection using the ssGBLUP method was higher than that using the traditional BLUP method (Table 3). From this result, the ssGBLUP method is a good replacement for the BLUP method in the Thai–Holstein dairy breeding program to increase genetic gain, reduce the generation interval, and select good genetic animals more precisely. However, one of the critical points that future research should improve upon is that increasing the reference population will lead to increased selection accuracy. These values were lower than those in Russian Holstein and black-and-white cattle (23%) [32] and in Nordic red dairy cattle (from 0.22 to 0.31%) [40]. Hayes et al. [24,25] reported that the accuracy of the genomic BLUP method was higher than that of the variable selection methods when using a small reference population, but it declined quickly when the relationship between the reference and candidate populations became weaker [24,25,41]. In another study, Daetwyler et al. [42] reported that the accuracies of the GEBVs ranged from 0.15 to 0.79 for wool traits in Merino sheep using a multibreed reference population with 7180 sheep. Misztal et al. [20] reported that the accuracy of the GEBVs was greater than that of the pedigree-based EBVs. At the same time, Sungkhareecha et al. [43] reported that the rate of genetic progress from the top 20% of the herd found that the ssGBLUP method was faster than the traditional BLUP method in milk traits, and it would be helpful for increasing the selection accuracy and reducing the costs of progeny testing. Although the accuracy of the GEBVs was already sufficiently high for dairy cattle, the adoption of the technology has been widespread, and future work will focus on improving this accuracy. This will be achieved by increasing the size of the reference populations used to derive the SNP prediction equations and by improving the density of the SNP markers to increase the LD. Using these results, we found that the GEBVs from the top 20% of the all animal, dam, and bull datasets were still greater than the EBVs, and using genotype data can increase the selection accuracy values. Therefore, the ssGBLUP method could improve the NSPC trait by increasing the selection accuracy, decreasing the generation interval, and reducing the costs of progeny testing, especially for traits with low heritability, such as fertility traits.

Our results presented in Table 4 showed that the GEBV values were more negative than the EBV values, indicating that the ssGBLUP method had better genetic values than the BLUP method, leading to faster genetic progress, especially in the GEBVs of the top 20% of the bull dataset, which had the highest values compared to the other datasets. This study had smaller GEBVs than in other populations, such as Russian black-and-white cattle populations, in which the average GEBV was 0.14 in cows and 0.05 in bulls [30]. This result indicated that the Thai–Holstein populations had better genetic values for the NSPC trait than the Russian black-and-white cattle populations. Within the breed group (BG), less than 87.5% Holstein genetics (BG1) had lower GEBVs than the other 87.5 to 93.6% Holstein genetics (BG2) and greater than the 93.7% Holstein genetics (BG3). One reason was that the Thai–Holstein populations with less than 87.5% Holstein genetics had a higher percentage of native cattle (Sahiwal, Brahman, Thai native) that were robust in the tropical environment. Therefore, the genetic values of dairy cows in this breed group did not change much compared to the other breed groups. In addition, heat stress in tropical areas is another risk factor that affects the NSPC values and their genetic values [5,44]. Additionally, Viana et al. [45] reported that the accuracy of GEBVs in low heritability traits was higher than average for the parents of young bulls without phenotype data. This could accelerate and improve the genetic gain for lowly heritable traits.

## 5. Conclusions

In conclusion, the animals with greater than 93.7% Holstein genetics had more services per conception than the animals with lower Holstein genetics. In addition to the factors other than genetic predisposition, it is likely this result also reflected the greater genetic adaptation of the crossbred cattle to the tropical environments in Thailand. At the same time, the single-step genomic BLUP method is a high-potential tool for improving fertility traits. Although that trait still has low heritability, it is possible to improve and increase the accuracy of the breeding value, and it can help determine an animal’s genetic progress. The accuracies in genomic selection depend on the number of reference populations, size of the genotype data, distribution, and contributions of genotypes and phenotypes to the genomic evaluation. Environmental and AI service management factors are important to the NSPC in small dairy farms in Thailand. In future research, it is recommended that the genomic estimated breeding value should be used where possible for multivariate production, reproduction, and possibly even adaptive traits such as the tolerance of the animals to high temperatures and humidity.

## Figures and Tables

**Figure 1 animals-13-03609-f001:**
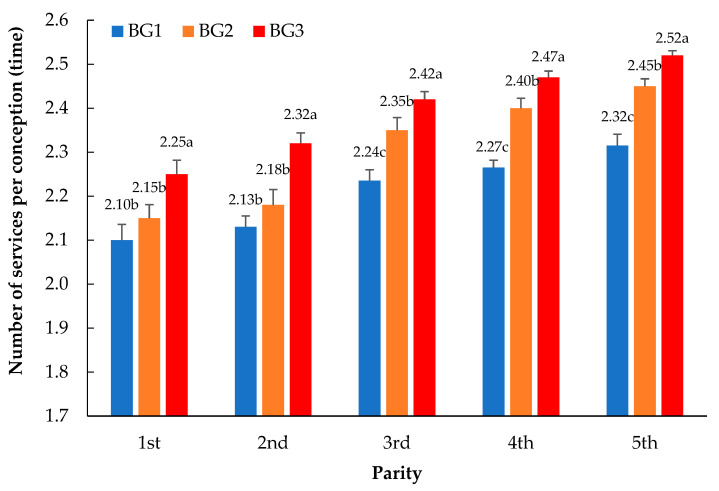
Comparison of the average number of services per conception (T; SE) separated by parity and breed group in Thai–Holstein crossbreds. The ^a,b,c^ superscripts indicate the significant differences (*p* < 0.05) in each breed group (BG1 = less than 87.5% Holstein genetics, BG2 = 87.5 to 93.6% Holstein genetics, and BG3 = greater than 93.7% Holstein genetics) within the parity groups.

**Table 1 animals-13-03609-t001:** The data structure of the number of services per conception trait for the estimation of the variance components, heritability, repeatability, and breeding values in Thai–Holstein crossbreds.

Items	Data Records			
Number of fixed effects				
Herd-year of insemination	1755			
Age group	7			
Breed group	3			
Parity	5			
Number of service sires (n)	995			
Number of animals with records (n)	54,027			
Number of animals in the pedigree (n)	109,233			
Number of animals with genotypes (n)	770			
**Breed groups (BG)** BG1 = less than 87.5%, BG2 = 87.5 to 93.6%, and BG3 = greater than 93.7%	Average	BG1	BG2	BG3
**Number of services per conception**				
Minimum (time)	1	1	1	1
Maximum (time)	10	10	10	10
Mean (time)	2.30	2.21	2.31	2.40
SD (time)	1.59	1.53	1.53	1.65
Number of records (n)	101,331	28,794	46,484	26,053

**Table 2 animals-13-03609-t002:** Averages of the posterior means (SE) of the variance components and the heritability and repeatability for the number of services per conception in Thai–Holstein crossbreds.

Parameters	Methods	
	Traditional BLUP	The ssGBLUP
$\sigma_{hy}^{2}$	0.16 (0.006)	0.16 (0.005)
$\sigma_{a}^{2}$	0.09 (0.010)	0.12 (0.012)
$\sigma_{p}^{2}$	0.26 (0.071)	0.25 (0.068)
$\sigma_{ss}^{2}$	0.02 (0.002)	0.02 (0.002)
$\sigma_{e}^{2}$	1.82 (0.025)	1.80 (0.022)
$h^{2}$	0.038 (0.002)	0.051 (0.002)
r	0.149 (0.014)	0.157 (0.015)
DIC	365,732.34	365,723.01
−2logL	50,000	48,852

$\sigma_{hy}^{2}$ = herd-year of insemination variance,$\;\sigma_{a}^{2}$ = additive genetic variance, $\sigma_{p}^{2}$ = permanent environmental variance, $\sigma_{ss}^{2}$ = service sire variance, $\sigma_{e}^{2}$ = residual variance, $h^{2}$ = heritability, r = repeatability, DIC = deviance information criterion, −2logL = negative two log-likelihood.

**Table 3 animals-13-03609-t003:** Comparison between the accuracy of the estimated breeding values (EBVs) from the BLUP method and the genomic estimated breeding values (GEBVs) from the ssGBLUP method for the number of services per conception separated by dataset.

Dataset	Traditional BLUP Method	The ssGBLUP Method	% Increase in Accuracy
All animals dataset	0.462	0.490	6.05
Dam dataset	0.465	0.498	7.10
Bull dataset	0.468	0.521	11.32
Top 20% of the all animals dataset	0.504	0.560	11.11
Top 20% of the dam dataset	0.508	0.572	12.60
Top 20% of the bull dataset	0.520	0.612	17.69

**Table 4 animals-13-03609-t004:** The average estimated breeding values (EBVs) from the BLUP method and the genomic estimated breeding values (GEBVs) from the ssGBLUP method for the number of services per conception separated by dataset and breed group.

Dataset	Breed Group	Traditional BLUP Method	The ssGBLUP Method
All animals dataset	BG1	−0.008	−0.009
	BG2	−0.008	−0.012
	BG3	−0.013	−0.018
Dam dataset	BG1	−0.008	−0.008
	BG2	−0.012	−0.019
	BG3	−0.022	−0.025
Bull dataset	BG1	−0.112	−0.118
	BG2	−0.116	−0.124
	BG3	−0.130	−0.136
Top 20% of the all animals dataset	BG1	−0.141	−0.149
	BG2	−0.150	−0.156
	BG3	−0.152	−0.159
Top 20% of the dam dataset	BG1	−0.157	−0.160
	BG2	−0.163	−0.166
	BG3	−0.169	−0.172
Top 20% of the bull dataset	BG1	−0.160	−0.165
	BG2	−0.172	−0.176
	BG3	−0.180	−0.185

BG1 = less than 87.5% Holstein genetics, BG2 = 87.5 to 93.6% Holstein genetics, and BG3 = greater than 93.7% Holstein genetics.

## Data Availability

Additional data are available upon request from the corresponding authors.

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
