# Peer review of "Genomic Selection Using Single-Step Genomic BLUP on the Number of Services per Conception Trait in Thai–Holstein Crossbreeds"

_animals, 2023, doi:10.3390/ani13233609_

Round 1

Reviewer 1 Report

Comments and Suggestions for Authors

The authors aimed to compare the genetic methods of the traditional BLUP and ssGBLUP techniques in terms of estimated genetic parameters and accuracy of estimated breeding values. They mentioned that the ssGBLUP method could improve the heritability value and accuracy of breeding values compared with the traditional BLUP method. Therefore, switching from traditional methods to the ssGBLUP method for the Thai dairy cattle breeding program is a viable option.

The Introduction section was written well.

The Materials described well.

The methods were well defined and understandable.

The Results section was written detailed and descriptive.

The Discussion section was written after detailed examination and comparable with the literature.

The Conclusion section was enough.

This manuscript can be accepted after minor revision.

1.       Simple Summary section should be written more popular language instead of scientific language.

2.       Keywords: “accuracy of selection” should be “accuracy”

3.       Line 119: Which post hoc test did you use?

Author Response

Dear Reviewer,

We answered questions and edited the manuscript based on suggestions from reviewer as follows:

Comments and Suggestions for Authors

Point 1: The authors aimed to compare the genetic methods of the traditional BLUP and ssGBLUP techniques in terms of estimated genetic parameters and accuracy of estimated breeding values. They mentioned that the ssGBLUP method could improve the heritability value and accuracy of breeding values compared with the traditional BLUP method. Therefore, switching from traditional methods to the ssGBLUP method for the Thai dairy cattle breeding program is a viable option.

 The Introduction section was written well.

The Materials described well.

The methods were well defined and understandable.

The Results section was written detailed and descriptive.

The Discussion section was written after detailed examination and comparable with the literature.

The Conclusion section was enough.

This manuscript can be accepted after minor revision.

Response 1: We thank you for your valuable time reviewing our manuscript and understanding the importance of this research.

Point 2: Simple Summary section should be written more popular language instead of scientific language.

Response 2: we have revised the simple summary according to the reviewer’s suggestion as follows: The development of dairy production systems has continued to progress, especially in purebred dairy cows in the United States and Europe. However, in-depth knowledge is still needed to support decision-making for crossbreed dairy cows raised in Asia to be used in breeding programs. The development of a new genetic method that is more effective than the current method is what everyone wants. Therefore, to ensure that the new method can replace the current method, it must be tested with the most confidence. See lines 18-23.

Point 3: Keywords: “accuracy of selection” should be “accuracy”

Response 3: The word ‘accuracy of selection’ is edited as ‘accuracy’. See line 43.

Point 4: Line 119: Which post hoc test did you use?

Response 4: We added the Scheffe’ (p<0.05) post hoc test in revised manuscript. See line 128.

Best Regards

Wuttigrai Boonkum

Reviewer 2 Report

Comments and Suggestions for Authors

This paper provides very timely and relevant information applicable to genetic improvement of dairy cattle (and other ruminants) farmed by smallholder farmers in tropical environments of the world, not only in terms of the methodologies that will yield best predictions of genetic merit, but also implies the potential need to consider breeding for tropical adaptation, either through use of greater crossbreeding with adapted breeds or for inclusion of traits such as heat tolerance in the breeding objectives of high grade Holstein cattle in the tropics. It is recommended that this paper be accepted for publication subject to relatively minor editorial-type changes shown below:

Line 27 - suggest replace ':" with a full stop and commence a new sentence with 'Therefore there is a need ...'

Line 32 - suggest insert 'was used to analyze the ...' after 'sampling'

Line 34 - the acronym for breed group (BG) has not been defined to this point so either spell out breed group in full or define the acronym as the percentage of Holstein genetics

Line 65 - delete 'in Thai-Holstein crossbreeds' as it is a repetition from the previous line

Line 71 - suggest replace 'they' with 'heritability' and change 'range' to 'ranges'

Line 98 - suggest insert 'being a successful conception' after "1"

Line 99 - suggest replace 'with' with 'being'

Line 106-108 - suggest include mention of the crosses involved in these breed groups in the materials and methods section, as currently mention of the types of crosses is delayed until the end of the discussion section (lines 325-326), meaning the reader is second-guessing interpretation of the results until the information is ultimately provided towards the end of the manuscript

Line 125 - suggest this line be changed to read '... modeled as constants.'

Lines 283-284 - suggest change the wording of these lines to '... effective estrus methods, artificial insemination at the right time, avoiding heat-stressed cattle, ensuring qualified artificial insemination technicians are used and they use effective hygiene before and after ...

Line 285 - it's not clear why the term 'bovine' is used here - should this be replaced with 'heifers' or some other term?

Line 330 - suggest the word 'traits' after 'heritability'

Line 331 - suggest replace 'low' with 'lowly'

Line 335 - in addition to 'factors other than genetic predisposition', it is likely this result also reflects the greater genetic adaptation of the crossbred cattle to the tropical environments in Thailand

Lines 341-342 - suggest the concluding sentence be strengthened to something along the lines of 'In future research, it is recommended that the genomic estimated breeding value should be used where possible for multivariate production, reproduction and possibly even adaptive traits such as tolerance of the animals to high temperatures and humidity.'

Comments on the Quality of English Language

See suggestions for minor editorial-type changes in the previous comments

Author Response

Dear Reviewer,

We answered questions and edited the manuscript based on suggestions from reviewer as follows:

Point 1: This paper provides very timely and relevant information applicable to genetic improvement of dairy cattle (and other ruminants) farmed by smallholder farmers in tropical environments of the world, not only in terms of the methodologies that will yield best predictions of genetic merit, but also implies the potential need to consider breeding for tropical adaptation, either through use of greater crossbreeding with adapted breeds or for inclusion of traits such as heat tolerance in the breeding objectives of high grade Holstein cattle in the tropics. It is recommended that this paper be accepted for publication subject to relatively minor editorial-type changes shown below:

Response 1: Thank you for your valuable time reviewing our manuscript and understanding the importance of this research.

Point 2: Line 27 - suggest replace ':" with a full stop and commence a new sentence with 'Therefore there is a need ...'

Response 2: We replaced ':" with a full stop and added the word 'Therefore there is a need ...'

 as you suggested. See line 25.

Point 3: Line 32 - suggest insert 'was used to analyze the ...' after 'sampling'

Response 3: We inserted 'was used to analyze the ...' after 'sampling'. See lines 30-31.

Point 4: Line 34 - the acronym for breed group (BG) has not been defined to this point so either spell out breed group in full or define the acronym as the percentage of Holstein genetics

Response 4: We already defined the full name of BG before using the abbreviation name, and we used the word Holstein genetics in the revised manuscript. See lines 32-34.

Point 5: Line 65 - delete 'in Thai-Holstein crossbreeds' as it is a repetition from the previous line

Response 5: We deleted 'in Thai-Holstein crossbreeds' as it is a repetition from the previous line. See lines 65-66.

Point 6: Line 71 - suggest replace 'they' with 'heritability' and change 'range' to 'ranges'

Response 6: We replaced 'they' with 'heritability' and changed 'range' to 'ranges' as you suggested. See line 72.

Point 7: Line 98 - suggest insert 'being a successful conception' after "1"

Response 7: We inserted 'being a successful conception' after "1". See line 107.

Point 8: Line 99 - suggest replace 'with' with 'being'

Response 8: We replaced 'with' with 'being'. See line 107.

Point 9: Line 106-108 - suggest include mention of the crosses involved in these breed groups in the materials and methods section, as currently mention of the types of crosses is delayed until the end of the discussion section (lines 325-326), meaning the reader is second-guessing interpretation of the results until the information is ultimately provided towards the end of the manuscript

Response 9: As you suggested, we added more details about Thai-Holstein crossbreeds in the introduction. See lines 53-55.

Point 10: Line 125 - suggest this line be changed to read '... modeled as constants.'

Response 10: We edited the word '... modeled as constants.'. See lines 133-134.

Point 11: Lines 283-284 - suggest change the wording of these lines to '... effective estrus methods, artificial insemination at the right time, avoiding heat-stressed cattle, ensuring qualified artificial insemination technicians are used and they use effective hygiene before and after ...

Response 11: The sentence is changed as you suggested. See lines 294-296.

Point 12: Line 285 - it's not clear why the term 'bovine' is used here - should this be replaced with 'heifers' or some other term?

Response 12: We replaced the word from 'bovine' with 'heifers'. See line 297.

Point 13: Line 330 - suggest the word 'traits' after 'heritability'

Response 13: We added the word 'traits' after 'heritability'. See line 342.

Point 14: Line 331 - suggest replace 'low' with 'lowly'

Response 14: We replaced 'low' with 'lowly'. See line 343.

Point 15: Line 335 - in addition to 'factors other than genetic predisposition', it is likely this result also reflects the greater genetic adaptation of the crossbred cattle to the tropical environments in Thailand

Response 15: The sentence is changed as you suggested. See lines 347-349.

Point 16: Lines 341-342 - suggest the concluding sentence be strengthened to something along the lines of 'In future research, it is recommended that the genomic estimated breeding value should be used where possible for multivariate production, reproduction and possibly even adaptive traits such as tolerance of the animals to high temperatures and humidity.'

Response 16: The sentence is changed as you suggested. See lines 356-359.

Point 17: Comments on the Quality of English Language

See suggestions for minor editorial-type changes in the previous comments

Response 17: we have carefully edited the entire MS to improve clarity as the reviewer’s suggestion.

Best Regards

Wuttigrai Boonkum

Reviewer 3 Report

Comments and Suggestions for Authors

Dear Editor and Authors,

I send you my review about the article Genomic selection using single-step genomic BLUP on the number of services per conception trait in Thai-Holstein crossbreds”.

The scope of the paper, as reported in the aim was to compare the genetic methods of the traditional BLUP and ssGBLUP techniques in terms of estimated genetic parameters and accuracy of estimated breeding values.

In my opinion, the paper result well structured and original, however, it need of some little change that I report below.

The introduction is well written and adequately to the aim of the research. However, in this chapter it should report some research that have studied similar aspects in the Holstein breed and in other breed similar to it.

The chapter Materials and methods is well structured and complete.

However, a briefly description of the BLUP method and of ssGBLUP methods should be reported.

The results is very well presented and they are very well discussed, also in comparison to the data reported in the literature. In addition, also the figures, are of quality and they well shown the data collected.

Nevertheless, in the chapter of materials and methods, of results and of discussion the authors largely use symbols like “<”, “>”.

In my opinion, to ease the understand the text by the readers, the symbols should be replaced with text, such as "less than.." or "greater than..."

Moreover, in the title of paragraph the acronyms should be avoid.

Finally, the conclusions resulted adequate to the data showed and to the aim of the research, but should be better highlighted the impact of the result of reserch.

Best regards

Author Response

Dear Reviewer,

We answered questions and edited the manuscript based on suggestions from the reviewer as follows:

Comments and Suggestions for Authors

Dear Editor and Authors,

I send you my review about the article Genomic selection using single-step genomic BLUP on the number of services per conception trait in Thai-Holstein crossbreds”.

The scope of the paper, as reported in the aim was to compare the genetic methods of the traditional BLUP and ssGBLUP techniques in terms of estimated genetic parameters and accuracy of estimated breeding values.

In my opinion, the paper result well structured and original, however, it need of some little change that I report below.

We are very grateful for the critical reading and your efforts to improve the quality of the manuscript.  We hope the responses to each comment as listed below will please you.

Point 1: The introduction is well written and adequately to the aim of the research. However, in this chapter it should report some research that have studied similar aspects in the Holstein breed and in other breed similar to it.

Response 1: We added more research of the Holstein breed and in other breed. See lines 73-77.

Point 2: The chapter Materials and methods is well structured and complete.

However, a briefly description of the BLUP method and of ssGBLUP methods should be reported.

Response 2: we added more information of the BLUP method and of ssGBLUP methods in revised manuscript as the reviewer’s suggestion. See lines 82-88.

Point 3: The results is very well presented and they are very well discussed, also in comparison to the data reported in the literature. In addition, also the figures, are of quality and they well shown the data collected.

Nevertheless, in the chapter of materials and methods, of results and of discussion the authors largely use symbols like “<”, “>”.

In my opinion, to ease the understand the text by the readers, the symbols should be replaced with text, such as "less than.." or "greater than..."

Response 3: We replaced with text ‘less than’ and ‘greater than’ for revised manuscript as you suggested. See revised MS.

Point 4: Moreover, in the title of paragraph the acronyms should be avoid.

Response 4: We edited the title of paragraph using the full name instead of acronyms as you suggested. See lines 126, 181, 212-213.

Point 5: Finally, the conclusions resulted adequate to the data showed and to the aim of the research, but should be better highlighted the impact of the result of research.

Response 5: We have edited the conclusion part as you suggested. See lines 347-349, and 356-359.

Best Regards

Wuttigrai Boonkum
